# Identification of circRNA-Associated-ceRNA Networks Involved in Milk Fat Metabolism under Heat Stress

**DOI:** 10.3390/ijms21114162

**Published:** 2020-06-11

**Authors:** Dongyang Wang, Zujing Chen, Xiaona Zhuang, Junyi Luo, Ting Chen, Qianyun Xi, Yongliang Zhang, Jiajie Sun

**Affiliations:** College of Animal Science, Guangdong Provincial Key Laboratory of Animal Nutrition Control, Guangdong Engineering & Research Center for Woody Fodder Plants, National Engineering Research Center for Breeding Swine Industry, South China Agricultural University, Guangzhou 510642, Guangdong, China; wangdy@stu.scau.edu.cn (D.W.); zujingchen@scau.edu.cn (Z.C.); zhuangxn@stu.scau.edu.cn (X.Z.); luojunyi@scau.edu.cn (J.L.); allinchen@scau.edu.cn (T.C.); xqy0228@scau.edu.cn (Q.X.)

**Keywords:** heat stress, milk fat, ceRNA, CD36

## Abstract

Summer temperatures are generally high in Southern China, and cows are likely to suffer a heat stress reaction. Heat stress will have a negative impact on the performance of dairy cows; however, the mechanism by which high temperature affects lactation is not clear. CircRNA is a type of non-coding RNA discovered in recent years, which performs a crucial function in many biological activities. However, the effects of circRNA on lactation function of dairy cows under heat stress is unknown. The present study aimed to explore the expression levels of circRNA in the mammary gland tissue of cows under heat stress. Firstly, we collected blood and milk samples of summer and winter cows and evaluated lactation performance using serum indicators, milk production, and milk composition. Incorporating the calculation of the temperature and humidity index, we conformed the heat stress status of cows in summer. Heat stress increased the concentration of HSP70 and decreased the concentration of SOD and PRL. Heat stress not only reduced milk yield but also affected milk quality, with milk lactose and milk protein decreasing with increased temperature. The analysis of the fatty acid composition in summer milk found significantly reduced concentrations of unsaturated fatty acids, especially long-chain unsaturated fatty acids. Sequencing of the cow’s mammary gland transcriptome revealed that compared to the appropriate temperature (ST) group, the heat stress (HS) group had a total of 2204 upregulated and 3501 downregulated transcripts. GO enrichment and KEGG pathway analysis showed that these genes were mainly related to milk fat metabolism. In addition, 19 upregulated and 19 downregulated circRNA candidates were found in response to heat stress. We used Pearson’s test to establish the correlation of circRNA-mRNA and identified four pairs of circRNA-miRNA networks between four circRNAs, six miRNAs, and the *CD36* gene. In this study, we revealed the possible role of circRNAs in lactation of dairy cows and identified that circRNA-miRNA-mRNA networks might exist in the cow’s mammary glands, providing valuable experience for dairy lactation and milk quality.

## 1. Introduction

As global temperatures rise [1], the hotter climate has become one of the main challenges for agriculture in this century, which affects the survival, growth, and reproduction of animals [2]. Their increased generation of metabolic heat means that high-yielding animals are extremely susceptible to heat stress in summer [3], which has caused huge economic losses to the livestock industry (about 2 billion per year in the United States) [4].

As one of the best foods for humans, milk has extremely high nutritional value. It not only contains most nutrients but also is a good source of many essential amino acids [5]. Dairy cows are intrinsically heat-tolerant; however, they can be susceptible to heat stress, which could result in significant decreases in milk production and milk quality [6]. Most studies have found that heat stress is associated with a decline in milk quality [7]. There has been a decrease in milk yield by 40% and in milk protein content by 4.8% in cows exposed to heat stress [8]. Compared with other seasons, milk protein content in summer reduces by 6% [7].

These large losses have resulted in extensive studies on the effects of heat stress on lactating cows [9]. According to the traditional concept, the reduced production performance of dairy cows during heat stress is closely related to decreased feed intake [10,11]. However, a series of recent studies have shown that the reduction in nutritional intake during heat stress accounts for only 35 to 50% of the reduction in milk synthesis [12,13]. Heat stress might directly affect milk production through unknown mechanisms unrelated to reduced intake, which might affect the expression of lactation genes by regulating hormone levels in cows [14]. Heat stress might increase the body’s gluconeogenesis, in which the catabolism of amino acids occurs to provide energy, and the supply of protein synthesis precursors is inadequate, which leads to a reduction in the content of milk protein [15,16].

Previously, changes in hormones [17] and their cellular levels [18] were believed to explain the effect of heat stress on the performance of dairy cows. Few studies have focused on the changes of ncRNA. CircRNA is a type of ncRNA discovered in recent years. It exists in large amounts in animals and plays an important role in the growth and development of animals [19,20]. CircRNA can act as miRNA sponges to regulate the transcription and splicing of parent genes and can be converted into peptides or proteins [21]. Increasing evidence shows that circRNA is involved in the most basic process of cells [22], and that heat stress can affect circRNA biosynthesis [23]; however, little is known about how circRNA regulates milk synthesis under heat stress. In the present study, we focused on the changes in circRNA levels of the mammary gland in lactating cows under heat stress and further explored the potential mechanism of circRNA-mediated regulation of milk synthesis in mammary tissue.

## 2. Results

### 2.1. The Performances of Dairy Cows under Heat Stress

We conducted the study during December 2018 and August 2019, respectively. In the two experimental groups, there was a difference between the THI index formed by ambient temperature and humidity (*p* < 0.01). The THI index in winter was significantly lower than 72, while the THI index in summer was significantly higher than 72 (Figure 1A). By measuring the physiological indicators of the cows, we observed that their respiration rate in summer was increased (*p* < 0.01; Figure 1B), and their rectal temperature in summer was higher than that in winter (*p* < 0.01; Figure 1B). The changes in stress-related hormone levels in the serum of dairy cows are shown in Figure 1C. The results showed that the serum level of HSP70 was higher in the heat stress (HS) group than in the appropriate temperature (ST) group (*p* < 0.01), and there was no significant change in IgG levels. However, the levels of SOD and PRL in the serum of dairy cows in summer decreased compared with those in winter (*p* < 0.05; Figure 1C). Table 1 shows the effect of the environment on milk performance. Compared with those in winter, in summer, the milk production of cows and the levels of milk proteins, fat, and lactose all decreased significantly, and somatic cell numbers in milk increased significantly. The fatty acid profile of milk showed that the levels of C18:1n9c fatty acids and unsaturated fatty acids were significantly decreased in summer compared with those in winter (Appendix A).

### 2.2. RNA Sequencing and Transcript Analysis

We constructed six cDNA libraries from cow mammary tissue collected during summer and winter, and each RNA-seq library produced approximately 110.69 ± 5.11 million raw reads. After quality correction, each library contained approximately 103.14 ± 4.69 million effective reads, accounting for approximately 93.19% of the original reads. We compared all valid reads to the *Bos taurus* reference genome and found that about 83.79 ± 1.24% of the valid reads could be successfully mapped to the genome, with 79.53 ± 1.82% proper pair alignments in the six libraries (Appendix A). These results proved the successful construction of this library and confirmed its suitability for the subsequent analysis of the sequencing results. Among the candidate transcripts from all the libraries, there were 153,575 unigenes in six mammary libraries, including 33.48% of the transcripts that completely matched the Ensembl known regions (Appendix A). For the annotated transcripts, we identified a total of 38,434 candidates, and 29,718 unigenes were expressed in all tissues (Appendix A). Additionally, *CSN1S1*, *CSN2*, *CSN1S2*, *PAEP*, *CSN3*, *LALBA*, *RPLP1*, *RPL37*, *EEF1A1*, and *COX1* were the top 10 highly expressed genes across the libraries, which are well-known as having key functions in the lactation process. We then performed principal component analysis (PCA) using the FPKM values of the identified transcripts (Figure 2A), and the results showed that the differences caused by environmental factors between the groups were much larger than those between the experimental individuals. To filter out false-positive results, we only retained transcripts that were expressed in at least three libraries. Finally, the study retained 20,277 tags for further analysis (Appendix A). Compared with the ST population, we identified 5705 significantly differently expressed transcripts, including 2204 upregulated transcripts and 3501 downregulated transcripts in HS libraries vs. the ST libraries (Figure 2B). GO enrichment analysis of the differentially expressed transcripts showed that these candidates were significantly enriched in the biological process of lipid metabolism-related functions (Appendix A), including ‘fatty acid beta-oxidation’, ‘fatty acid biosynthetic process’, ‘fatty acid beta-oxidation using acyl-CoA dehydrogenase’, ‘fatty acid metabolic process’, ‘lipid transport’, and ‘lipid homeostasis’ (Figure 2C). Among the differentially expressed transcripts, we identified 5428 genes that participated in 77 KEGG signaling pathways (Appendix A), including ‘prolactin signaling pathway’, ‘fatty acid metabolism’, ‘biosynthesis of unsaturated fatty acids’, and ‘PPAR signaling pathway’ (Figure 2C,D). Thus, the GO enrichment and KEGG pathway analysis provided some hints for the variabilities of milk fat content in cows under heat stress.

### 2.3. Identification of circRNAs

Based on anti-splicing reads generated from high-throughput RNA sequencing datasets, researchers have developed several tools to identify circular RNAs. In this paper, we used five different circular RNA prediction algorithms, which identified 61,175 unique circRNAs in the six libraries. A comparison of the prediction results demonstrated differences in the outputs of the five algorithms. Find_circ predicted the most circRNA types (*n* = 44,330), and MapSplice predicted the fewest (*n* = 7719). CIRCexplorer2, CircRNA_finder, and CIRI identified 27,565, 38,358, and 15,220 circRNAs, respectively (Appendix A). To eliminate the possible errors in the algorithms, further analyses only used candidate circRNAs predicted by all five algorithms (*n* = 2950) (Figure 3A). These circularization events were produced from 1350 host loci, including 1600 transcripts that generated more than one circRNA (Appendix A). Of the 2950 identified circRNAs, only 38 were differentially expressed between ST and HS individuals. Compared with the ST group, 19 circRNA candidates were upregulated, and 19 were downregulated (Figure 3B).

### 2.4. Functional Interactions between circRNAs and mRNAs

To discover the interactions between circRNA and mRNA expression, we firstly performed Pearson’s tests on the 38 differentially expressed circRNAs and 29 PPAR pathway genes (Figure 2D). This analysis identified 255 significantly positively correlated circRNA-mRNA interactions with *p* < 0.05 and R > 0.8 (Appendix A). In general, miRNAs play an important role in the processes of circRNA and mRNA interactions [21]. Among data deposited at the NCBI, we collected 23 bovine mammary gland libraries and identified 861 miRNAs (Appendix A). Next, we selected the 242 mammary-enriched miRNA candidates that were expressed in all libraries for further analysis (Appendix A). We then evaluated the putative interactions between differentially expressed circRNAs and PPAR pathway-related genes from this study for competitive binding with shared mammary-special miRNAs using the miRanda algorithm (http://www.microrna.org/microrna/home.do), with energies of ≤ −20.0 kcal/mol and no mismatch in the seed region. The analysis identified 427 circRNA-miRNA interactions (Appendix A) and 732 mRNA-miRNA interactions (Appendix A). Subsequently, we constructed 890 circRNA-miRNA-mRNA networks (Appendix A). Generally, almost all long-chain fatty acids are considered to be derived from diet digestion and absorption [24], and the expression levels of cluster of differentiation 36 (CD36) and solute carrier family 27 (SLC27) are stimulated for fatty acid uptake and transport during lactation [25]. In the present study, the expression levels of *CD36* and *SLC27A6* decreased significantly in response to heat stress. Moreover, *CD36* expression was significantly and positively associated with the expression of 14 different circRNA candidates, while *SLC27A6* was associated with eight candidates (Appendix A). We constructed circRNA-miRNA-CD36 (SLC27A6) networks by pairing the shared miRNA recognition sequences. In detail, circRNA-miRNA-CD36 included 10 circRNAs (*circZCCHC17*, *circFCHSD2*, *circCD36_2*, *circMAP7*, *circKANSL1*, *circHNRNPLL*, *circCD36*, *circHNRNPLL_1*, *circRASEF*, and *circLARP1B*) and 16 miRNAs (miR-450b, miR-500, miR-6516, miR-2284a, miR-2284g, miR-2284y, miR-2285av, miR-2285cf, miR-2285cp, miR-2285cr, miR-2285db, miR-11986b, miR-345-3p, miR-502b, miR-218, and miR-378b), and the circRNA-miRNA-SLC27A6 network only contained circCSN1S2 and miR-223 (Appendix A).

### 2.5. Characterization of circRNA

To prove the circular structure of circRNA, we selected five highly expressed candidates (*circACACA*, *circCSN1S1*, *circEZH2*, *circPRKAA2*, and *circTMEM120B_2*) and 11 unique circRNAs that were positively correlated with *CD36* and *SLC27A6* expression for further analysis. We designed a pair of divergent primers and convergent primers and then used cDNA or genomic DNA (gDNA) as a template for amplification. Generally, amplification was successful for both cDNA and gDNA templates using the convergent primers for the tested circRNAs. However, amplification using the divergent primers was successful for only nine candidates in cDNA but not genomic DNA (Figure 4A). We verified the junction sequences of the nine putative circRNAs using Sanger sequencing (Appendix A). Next, quantitative real-time reverse transcription PCR (qRT-PCR) analysis of the nine circRNAs and their parent genes was carried out, and we found that the expression levels of the circRNA candidates were not significantly changed between RNase R treated and mock groups. However, the mRNA expression levels of the host genes and *GAPDH* were different (Additional file 10B; Figure 4B). In addition, the results showed that these nine circRNA candidates were consistent between RT-qPCR and sequencing analysis between ST and HS groups (Appendix A), suggesting that our estimation of abundance was accurate. In detail, the expression of these circRNAs was significantly downregulated in the HS group, except for *circKANSL1* and *circTMEM120B_2*. These two circRNAs showed to descend in the HS group, but no significant levels were detected.

## 3. Discussion

As a typical heat-resistant animal, the appropriate feeding temperature for dairy cows is 5–25 °C. During the summer in southern China, the temperature is much higher, and cows are extremely sensitive to changes in warm environments because of their high metabolic rate. Therefore, cows are particularly vulnerable to heat stress [13]. Numerous studies have shown that THI can effectively identify whether cows are under heat stress. Usually, THI = 72 is used as a threshold for judging the heat stress status of cows. When THI is higher than 72, cows develop heat stress [26]. In the present study, the THI index in summer was higher than 72, indicating that the cows were in the state of heat stress. The THI index was lower than 72 in winter, and the cows were in good condition. As a constant temperature animal, cows need to control their body temperature within a certain range to maintain normal metabolism; however, the balance of the body is disturbed by external factors, such as heat stress. When cows are in a heat stress state, the main way to reduce the adverse effects of heat stress is to accelerate heat dissipation to maintain their heat balance [27]. Cows increase heat dissipation mainly by increasing their respiration rate. When the heat cannot be dissipated completely, the cows’ rectal temperature will increase [28]. Therefore, rectal temperature and the respiratory rate can be used as direct indicators to evaluate heat stress in dairy cows [29]. In the present study, the respiratory rate and rectal temperature in summer were significantly higher than those in winter, confirming that the cows were suffering from heat stress in summer.

Heat stress proteins exist widely in various tissues, and their expression level is low under normal conditions [30]. As an important molecular chaperone, HSP70 has anti-apoptosis, anti-oxidation, cell proliferation promotion, and immune regulatory functions [31]. When heat stress occurs, HSP70 can protect its tissue cells from the damage caused by adverse stress stimulation by increasing its synthesis, thus playing an important role in stress tolerance and protection [32]. Previous studies have found that heat stress can increase HSP70 levels in goats [33], chickens [34], and fruit flies [35], which is similar to the results in the present study. When the human body is in a state of heat stress, cell metabolism is accelerated, respiratory function is enhanced, and a large amount of reactive oxygen species is produced in the body [36]. Under normal circumstances, the free radicals metabolized by the body will be neutralized by the antioxidant system [37]. However, previous studies have found that heat stress can reduce the antioxidant capacity of the body, generating oxidative stress [38,39] and reducing immune function [40,41]. These results are consistent with the results of SOD reduction in this experiment. However, there was no significant change in IgG levels in this experiment, which might be caused by the self-regulation of chronic heat stress. Studies have found that heat stress reduces the serum concentration of PRL in dairy cows [42]. PRL is essential in mammary development and lactation [43]. Consistent with previous studies, our data showed that PRL levels in dairy cows decreased significantly under heat stress.

In the present study, we found that heat stress had a significant effect on milk yield. The output per cow in winter was as high as 22.83 ± 0.2251 kg/day, while during the heat stress period in summer, this was reduced markedly to 19.06 ± 0.1643 kg/day. Ravagnolo and Misztal [44] and Kadzere et al. [3] reported similar results. Under heat stress, cows’ milk yield decreases by 10–30% [45]. Habeeba et al. analyzed the milk production of cows at ambient temperatures of 20, 30, and 40 °C and found that the milk yield at 30 and 40 °C decreased significantly by 14.7% and 55%, respectively [46]. Heat stress not only affects the milk yield of dairy cows but also changes its composition. With the increased temperature, the contents of milk yield and milk protein of dairy cows decrease rapidly [47]. Beede and Sheare found a negative correlation between milk fat levels and ambient temperature [48]. Goat milk has shown a similar trend, with a significant negative correlation between ambient temperature and the milk’s physicochemical properties [49]. With the increase of ambient temperature, our study found that cows were in a state of heat stress and the levels of milk lactose, fat, and protein all declined, which was similar to the results of previous studies [50,51,52]. Excessive oxygen free radicals produced by the body during heat stress easily oxidize unsaturated fatty acids (especially polyunsaturated fatty acids); therefore, the content of saturated fatty acids in milk increases, while the content of unsaturated fatty acids decreases [53]. Our study found similar results, and in pig milk, Christon et al. found that high temperatures could negatively affect the metabolism of polyunsaturated fatty acids [54].

A previous study has found that under heat stress, the dry matter intake and milk production of dairy cows are correlated significantly and negatively with THI [55,56]. Energy intake changes during heat stress have long been considered to be responsible for the decrease in milk quality during heat stress. However, recent studies have shown that the reduction of nutrient intake accounts for only 35–50% of the reduction in heat-induced milk synthesis [17]. The remainder might be caused by the non-matching of genetically-regulated nutrients in the body during heat stress [57]. Therefore, in the current study, we used RNA-Seq to investigate the potential molecular mechanisms underlying the changes in cow milk synthesis under heat stress. The sequencing results identified 38,431 unique Ensembl genes that were expressed in the mammary tissue of lactating dairy cows, among which lactation-related genes were highly expressed. Similar results were obtained previously in cows [58], goats [59], and humans [60]. In addition, we focused on identifying differentially expressed genes in response to high ambient temperature. GO analysis indicated that these genes participated in the biological process of milk fat metabolism, but their expression levels decreased with the onset of heat stress. KEGG analysis showed that differentially expressed genes, such as *CD36*, *PPARG*, *SCD*, and SLC27, were rich in PPAR signaling pathways and played an important role in fatty acid absorption, de novo synthesis, triglyceride synthesis, and milk fat droplet secretion [61]. In many mammals, mammary tissue is one of the organs with the strongest fatty acid metabolism [62]. Generally, fatty acid metabolism in the mammary gland involves a more complex network [63]. Among the above-mentioned factors, CD36, as an important fatty acid uptake protein, shows a gradually increasing expression level during lactation, reaching its peak at the zenith of lactation [25]. PPARG is a core factor that regulates fatty acid metabolism during lactation [25]. SCD is a protease bound to the endoplasmic reticulum, which catalyzes saturated fatty acids to unsaturated fatty acids, thereby directly affecting the composition of fatty acids in milk [64]. Through the identification of dairy cow transcripts, we found that *SLC27* expression was the highest among genes related to fatty acid metabolism [25]. In the present study, changes in the composition of fatty acids in milk also verified the roles of these genes in milk production under heat stress.

CircRNAs, as a type of non-coding RNAs that exist widely in mammals, have attracted increased attention recently [65]. Many studies have shown that circRNAs can act as competitive endogenous RNAs (ceRNAs) to regulate miRNAs and mRNAs, thereby exerting their biological functions [66]. Recently, a study on the mammary gland of dairy cows has shown that a circRNA derived from the *CSN1S1* gene is highly expressed in the mammary gland, contains many miR-2284 family binding sites, and might serve as a miR-2284 sponge to regulate casein translation [67]. To explore the regulatory effect of circRNAs on lactation of dairy cows under heat stress, we performed a circRNAome analysis on the mammary tissue of cows between the ST and HS groups, in which five algorithms identified 2950 circRNAs, 38 of which were differentially expressed. Pearson’s test was used to identify circRNA-mRNA correlations, which identified 255 positive correlations. The expression levels of *CD36* and *SLC27A6* genes have extremely important effects on the uptake of long-chain fatty acids [68]; therefore, based on the ceRNA theory, we focused on the circRNA candidates positively related to *CD36* and *SLC27A6*. Considering their covalent closed-loop structure [69], we finally identified 11 circRNAs through the design of divergent and convergent primers, Rnase R digestion, and Sanger sequencing. Next, we performed a Pearson correlation analysis on the circRNA-miRNA and miRNA-mRNA interactions. Based on their shared miRNA regulatory elements, we found four pairs of circRNA-miRNA-mRNA networks: circFCHSD2-miR6516-CD36; circHNRNPLL-miR11986b-CD36; circKANSL1- miR345 and miR502b and miR6516-CD36; circMAP7-miR11986b and miR345-CD36. These four pairs of circRNA-miRNA-mRNA might play an important role in the milk fat metabolism of dairy cows under heat stress. The construction of the ceRNA network provides a new direction for research into bovine lactation and provides a theoretical basis for improving the performance of cow milk production in the future.

In summary, we found that heat stress, which has brought huge economic losses to the dairy industry, had a negative effect on milk quality with increasing ambient temperature. Heat stress markedly altered the expression patterns of circRNAs in dairy cow’s mammary gland tissue. These heat-induced circRNAs participated in the regulation of milk fat metabolism through ceRNA networks. Our results provided a new theoretical basis to study the function of circRNAs in dairy cows in response to heat stress.

## 4. Materials and methods

### 4.1. Experimental Design and Sample Collection

We conducted the experiment at a dairy farm in Guangzhou, Guangdong Province, China. We selected 60 cows balanced for their lactation stage and parity and divided them into two groups, each group containing 30 animals. One group was assessed during December 2018 (the appropriate temperature (ST) group). The other group was assessed in the summer (August) 2019 and termed the heat stress (HS) group. Appendix A showed the composition and nutritional level of the feed. All cows were fed the same commercial formula diet and raised under the same management conditions. We obtained the temperature and relative humidity data of the cowshed at 14:00 and 22:00 every day using five thermometers and hygrometers placed in the center and around the cowshed to calculate the THI. In the course of the experiment, we measured the rectal temperature and respiration rate of each cow from 10:00 to 11:30 and 16:00 to 17:30 on two consecutive days each week. We obtained the rectal temperature by leaving the animal thermometer in the rectum for 3 min. The breathing rate was based on the up and down movement of the chest and abdomen within 1 min by visual inspection. During the test period, each cow’s milk production was recorded daily. We collected fresh milk samples every 10 days and used to determine the milk composition. Three days before the end of the experiment, we collected blood samples via tail vein blood collection, and we detected the concentrations of SOD, IgG, HSP70, and PRL in serum using ELISA kits (Nanjing Jiancheng Bioengineering Institute, Nanjing, Jiangsu, China).

In each group, six animals that were balanced for milk yield were chosen randomly, and percutaneous biopsies from each cow were obtained from the right rear quarter of the mammary gland. The mammary glands were washed with diethyl pyrocarbonate (DEPC)-treated PBS, excess tissue and fat were removed, and the glands were cut into small pieces, placed in cryopreservation tubes, marked, and stored in liquid nitrogen as samples for RNA extraction. All procedures were performed in accordance with the procedures approved by the Institutional Animal Care and Use Committee of South China Agricultural University (Ethics Approval Code: SCAU2018F006, 13 March 2018).

### 4.2. RNA Preparation and Sequencing

In accordance with the manufacturer’s instructions, we used the Trizol reagent (Invitrogen, Carlsbad, CA, USA) to extract the total RNA from the mammary tissue. To remove genomic DNA from each RNA sample, we used DNase I (Takara, Dalian, Liaoning, China). Only RNA samples with suitable RNA electrophoresis results (28S/18S ≥ 1.0) and RNA integrity number (RIN) ≥ 7.5 could be analyzed further. In each experimental group, we used 5 μg of each RNA sample and mixed in pairs to prepare a total of three RNA pools. A Ribo-Zero^TM^ rRNA Removal Kit (Illumina, San Diego, CA, USA) was used to remove the ribosomal RNA from the RNA libraries, and the remaining RNA fragments were reverse transcribed using an RNA-seq Library Preparation Kit (Illumina) to form the final cDNA. Finally, we used an Illumina Hiseq 4000 sequencer (LC Bio, Hangzhou, Zhejiang, China) for paired-end sequencing of the libraries.

### 4.3. Bioinformatic Analysis

We first used FastQC v0.11.9 (http://www.bioinformatics.babraham.ac.uk/projects/fastqc/) to quickly evaluate the quality of the raw sequencing data to ensure the accuracy of subsequent analysis. Then, we used Cutadapt to filter the data, which can remove the reads that contain low-quality data and bases contaminated by the Adapter v2.10 [70]. We compared the filtered data with the *Bos taurus* reference genome downloaded from the Ensembl Genome website using TopHat v2.1.1 (http://ccb.jhu.edu/software/tophat/index.shtml). We then used Cufflinks v2.2.1 (http://cole-trapnell-lab.github.io/cufflinks/cuffdiff/index.html) to assemble the transcripts of the TopHat alignments, estimate the abundance of target transcripts, and detect the differential expression between samples [71]. In this study, we used five algorithms to predict circRNA: MapSplice [72], find_circ [73], circRNA_Finder [74], CIRI [75], and CIRCexplorer2 [76]. To eliminate the possible errors in the algorithm, only circRNA candidates identified by all five algorithms were processed for further analysis. After the above preliminary analysis, we evaluated the expression level of all coding genes in the library using FPKM and calculated the expression abundance of circRNA using the number of splice junctions [77]. Then, we used Cuffdiff v2.2.1 [71] to compare the expression levels of coding transcripts and used the edgeR package v3.11 [78] to calculate the differentially expressed circRNA between the ST and the HS groups, with a q value of < 0.05. Finally, to evaluate the main functional pathways of the differentially expressed genes, the GO terms and KEGG pathway analysis were performed using DAVID gene functional classification (https://david.ncifcrf.gov/), and obvious enrichment results were obtained with *p* values < 0.05. In addition, we downloaded 23 miRNA-seq data sets for bovine mammary glands from the gene expression omnibus (GEO) database (https://www.ncbi.nlm.nih.gov/gds/) with the accession numbers PRJNA542496, PRJNA482122, and PRJNA248657. We then used the miRDeep2 program [79] to further analyze the miRNA libraries by BLAST searching against bovine miRBase annotation (http://www.mirbase.org/) to identify mammary-specific miRNA candidates.

### 4.4. Validation of circRNAs

In accordance with the manufacturer’s instructions, we used Trizol reagent (Invitrogen) to extract the total RNA from mammary tissues. Subsequently, we used a PrimeScript RT Reagent Kit with gDNA Eraser (Takara, Dalian, Liaoning, China) to reverse transcribe RNA to cDNA. To verify the circular structure of the circRNA, we designed a pair of convergent and divergent primers and verified their head-to-tail splicing using PCR and Sanger sequencing. Then, we performed RNase R treatment, in which 2 μg of total RNA was incubated with 3 U/μg RNase R at 37 °C for 10 min. As a control, another 2 μg total RNA was incubated with RNase-free water under the same conditions. After treatment with RNase R, we detected the RNA expression levels of circRNAs and their linear mRNA using quantitative real-time PCR (qPCR). We used *GAPDH* (glyceraldehyde 3-phosphate dehydrogenase) as an internal reference for data analysis. Appendix A shows the primer sequences used in the present study.

## Figures and Tables

**Figure 1 ijms-21-04162-f001:**
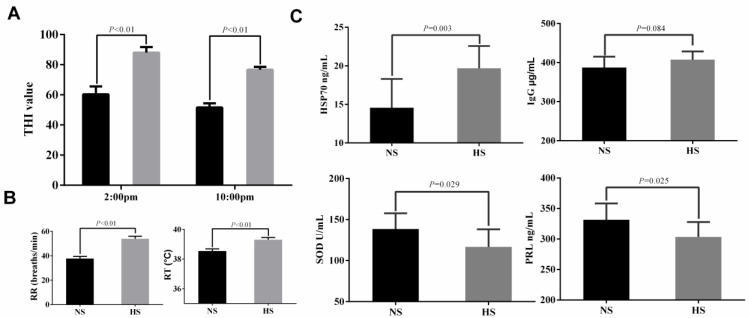
Heat stress-related indicators were identified during the experimental period. (**A**) changes in the temperature-humidity index at different times; (**B**) respiratory rate and rectal temperature of cows; (**C**) analysis of serum biochemical indices between different groups. ST, the appropriate temperature group; HS, heat stress group. Black histograms represent NS, and grey histograms represent HS.

**Figure 2 ijms-21-04162-f002:**
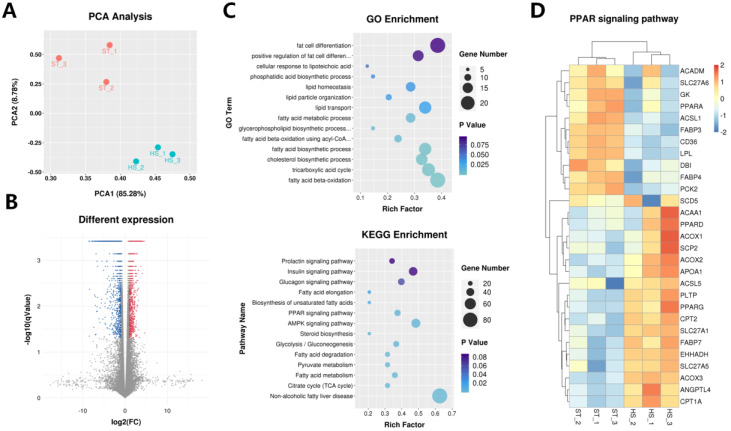
Transcriptome analysis of the mammary gland of dairy cows under heat stress. (**A**) principal component analysis of the whole transcripts in the mammary gland of dairy cows under heat stress; (**B**) volcano plot, showing significantly differentially expressed genes; (**C**) GO and KEGG analysis of differentially expressed genes; (**D**) heatmap, displaying the lactation-related gene identified in PPAR signaling pathway. ST, the appropriate temperature group; HS, heat stress group.

**Figure 3 ijms-21-04162-f003:**
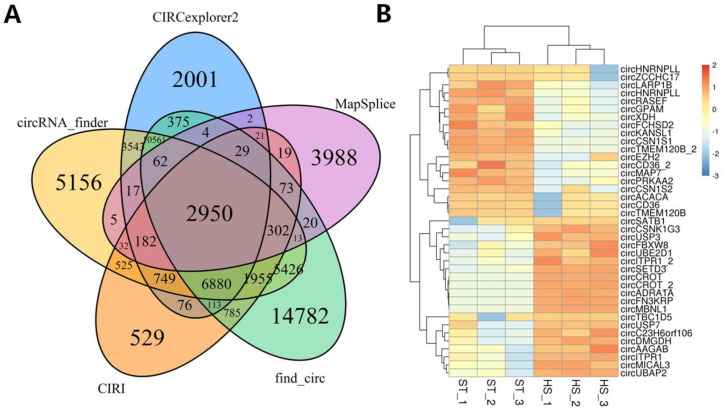
CircRNAome analysis of the mammary gland of dairy cows under heat stress. (**A**) Common circRNA candidates identified by all five approaches; (**B**) Heatmap, showing significantly differentially expressed circRNA candidates. ST, the appropriate temperature group; HS, heat stress group.

**Figure 4 ijms-21-04162-f004:**
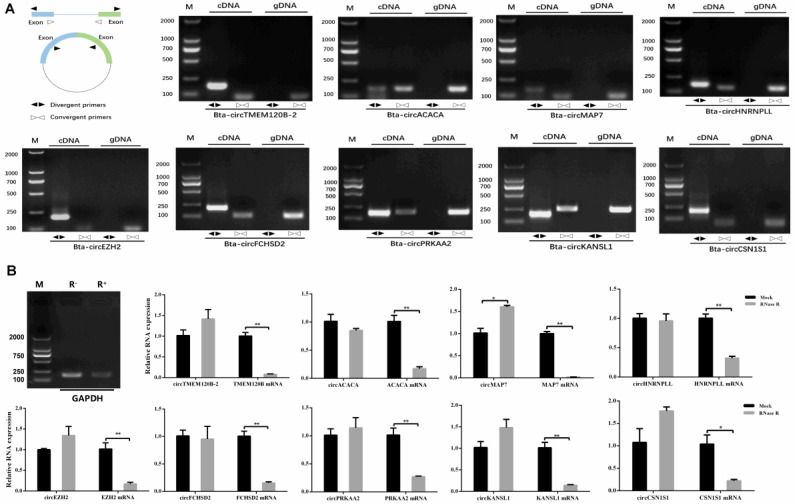
Characterization of circRNA candidates. (**A**) identification of circRNA candidates by divergent and convergent primers; Divergent primers amplify circRNA candidates in cDNA but not genomic DNA, and convergent primers amplify targets in both cDNA and genomic DNA. (**B**) RT-qPCR for the abundance of circRNA and their hosting mRNA treated with RNase R.

**Table 1 ijms-21-04162-t001:** Milk yield and composition in dairy cows.

Item	NS	HS
Milk yield kg/day	22.83 ± 0.2251 ^A^	19.06 ± 0.1643 ^B^
Protein %	3.46 ± 0.0336 ^A^	3.30 ± 0.0329 ^B^
Fat %	3.71 ±0.0352 ^A^	3.55 ± 0.0253 ^B^
Lactose %	4.69 ± 0.0292 ^a^	4.58 ± 0.0318 ^b^
Somatic cell counts 10^4^/mL	30.47 ± 0.5833 ^B^	37.83 ± 0.7445 ^A^

ST, the appropriate temperature group; HS, heat stress group. a and b denote values that differ significantly at *p* < 0.05, and A and B denote values that differ significantly at *p* < 0.01.

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
