# Peer review of "Identification of circRNA-Associated-ceRNA Networks Involved in Milk Fat Metabolism under Heat Stress"

_ijms, 2020, doi:10.3390/ijms21114162_

Round 1
Reviewer 1 Report
I think the article is very interesting and well explained.
Nevertheless, I have some doubts about the methodology I would like to discuss it with the authors, to infer how the resuts can be affected.
- Something very important to note is that TopHat 2 is quite old and it contains several bugs that have been fixed in other softwares such as hisat2. In fact, the developer of this software is quite clear about this:
https://twitter.com/lpachter/status/937055346987712512
- On page 3 line 20/21 you mention: 5707 significantly differentially expressed transcripts ... FDR <0.05? any FC cut off? It is a bit anoing tomove to methods to check what cut-off values you have used, please could you include it in the results to facilitate reading ?
- On page 3 line 48 you mention significantly positively correlated ... what does it mean? P <0.05 R> 0, R> 0.8?
- You are include libraries from other authors to see how the expression of miRNAs is. In a great majority of cases, in a normal RNASeq you can also check the expression of miRNAs, without having to use data from miRNASeq. Have you checked if this is so? Or if there is similarity between the miRNAs of the miRNASeq and the RNASeq?
- On page 7, line 43/44 you talk about RPKMs when you previously talked about FPKMs. Why are you using two types of measurement. I don't know if it makes sense to mix cufflinks / FPKM with edgeR / RPKM when edgeR can fully do the whole analysis.
- What tool has been used for functional enrichment?
- Could you explain why you have not used the same cows in December / August and do a paired study?
Author Response
Comments and Suggestions for Authors
(1) I think the article is very interesting and well explained.
Nevertheless, I have some doubts about the methodology, and I would like to discuss it with the authors, to infer how the resuts can be affected.
Answer: Many thanks for your kind comments and suggestions.
(2) Something very important to note is that TopHat 2 is quite old and it contains several bugs that have been fixed in other softwares such as hisat2. In fact, the developer of this software is quite clear about this: https://twitter.com/lpachter/status/937055346987712512
Answer: Yes, HISAT2 is a fast and sensitive alignment program for mapping next-generation sequencing reads, and it was an updated version and shared core modules of TopHat2. However, our softwares did not updated in time, and we did not do well in HISAT2 analysis at that time. In the future, we will use the HISAT2 algorithm to do our next mapping. Thanks so much for your good suggestion.
(3) On page 3 line 20/21 you mention: 5707 significantly differentially expressed transcripts. FDR <0.05? any FC cut off? It is a bit annoying to move to methods to check what cut-off values you have used, please could you include it in the results to facilitate reading?
Answer: In our manuscript, we used Cuffdiff to compare the expression levels of coding transcripts, and this protocol reveals markedly differentially expressed genes and transcripts between two or more conditions according to the false discovery rate of 5% (Trapnell et al., 2012). We therefore only applied FDR value and no FC cut-off to calculate the different expression based on the publication of Trapnell et al.
Trapnell C, Roberts A, Goff L A, et al. Differential gene and transcript expression analysis of RNA-seq experiments with TopHat and Cufflinks. Nature Protocols, 2012, 7(3): 562-578.
(4) On page 3 line 48 you mention significantly positively correlated ... what does it mean? P <0.05 R> 0, R> 0.8?
Answer: Significantly positively correlation between circRNA-mRNA interactions were analyzed with P <0.05 and R> 0.8, and we have revised in main text with new line 167. Thanks so much.
(5) You are include libraries from other authors to see how the expression of miRNAs is. In a great majority of cases, in a normal RNASeq you can also check the expression of miRNAs, without having to use data from miRNASeq. Have you checked if this is so? Or if there is similarity between the miRNAs of the miRNASeq and the RNASeq?
Answer: In our paper, we remove the ribosomal RNA and subsequently constructed RNA libraries for RNA-seq to find coding genes and circRNA candidates. However, we did not enriched approximately 25-bp fragments to prepare miRNA-seq libraries, and the miRNA libraries from other papers were used to identify mammary-specific miRNA candidates. In 23 bovine mammary gland libraries of miRNA-seq, 861 miRNA candidates were identified and 242 mammary-enriched miRNA candidates that were expressed in all libraries for further analysis.
(6) On page 7, line 43/44 you talk about RPKMs when you previously talked about FPKMs. Why are you using two types of measurement. I don't know if it makes sense to mix cufflinks / FPKM with edgeR / RPKM when edgeR can fully do the whole analysis.
Answer: we do apologized for our mistake, and we have revised “RPKMs” to “FPKMs” in the main text with new line 402.
In our paper, we used Cuffdiff to compare the expression levels of coding transcripts with FPKM values. And we used the edgeR package to calculate the differentially expressed circRNA with the number of junction reads between the ST and the HS groups. The details were showed in main text with new line 401-405 and Additional file 7.
(7) What tool has been used for functional enrichment?
Answer: Biological Processes GO terms and KEGG pathway analysis were performed using DAVID gene functional classification (https://david.ncifcrf.gov/) in our paper, and we have added the information in new line 406-409. Thanks so much.
(8) Could you explain why you have not used the same cows in December / August and do a paired study?
Answer: Thanks so much for your comments. In our paper, the tested cows were balanced for their lactation stage and parity. In December 2018/ August 2019, the same cows were at different stages of lactation.
Reviewer 2 Report
Wang and colleagues investigate the role of heat stress on milk production in dairy cows. Heat stress reduces milk production and affects milk quality. Analyses of the fatty acid composition in the milk produced during summer display reduced concentrations of unsaturated fatty acids, especially long-chain unsaturated fatty acids. The authors performed RNA sequencing experiments and constructed six cDNA libraries from cow mammary tissue collected respectively during summer and winter. GO enrichment and KEGG pathway analysis showed that affected genes were mainly related to milk fat metabolism. Analysis of circRNAs revealed 19 upregulated and 19 downregulated circRNAs modulated in response to heat stress. Pearson’s test was used to establish the correlation of circRNA-mRNA and allowed identification of putative networks linking circRNAs-miRNAs to CD36 and/or SLC27. The results are interesting. However, the reported findings remain quite elusive and speculative and need to be validated by a series of experimental procedures. It is strongly recommended to include new data obtained with validation experiments of the identified molecular axes in the revised manuscript. Some additional files are either absent or impossible to open. Please double check.
Overall, the authors need to provide more complete and rigorous data to support their claims.
Additional points are listed below
- 2 Line 24-25. The reference number 21 is not appropriate. The manuscript from Poliseno and colleagues discloses a novel mechanism of PTEN regulation which is mediated by the sponge activity of PTENP1 toward specific microRNAs. CircRNAs are not mentioned in this paper.
- 3 Line 49-50. Please include a relevant reference for the statement “In general, miRNAs play an important role in the processes of circRNA and mRNA interaction”. Why did the authors decide to focus on the putative role of microRNAs as mediators of the axis circRNAs-mRNAs? CircRNAs can act as protein sponges or even be translated into small peptides (Kristensen et al., 2019).
- 4 Line 13. The Authors should provide a graph related to the downregulation of CD36 and SLC27 after HS.
- The expression levels of the validated circRNAs need to be assessed in the two experimental conditions used for the RNA-seq experiments (normal temperature and heat stress).
- The authors focused on the miRNA-sponging function of the identified circRNAs. The authors should ascertain the levels of the target miRNAs and their relative mRNA targets in the mammary glands in the two experimental conditions (normal temperature and heat stress), in order to validate the circRNA-miRNA-mRNA networks mentioned in the discussion.
Minor points:
- In Figure 1A/1B/1C, a detailed legend would help in distinguishing the two experimental conditions, by indicating which colors the histograms are referred to.
- Additional file 4 – impossible to open.
- Additional file 5 and 6 are not present in the provided supplementary material.
- Figure 2D should be positioned after figure 3B, coherently with the order of the main text.
Author Response
Wang and colleagues investigate the role of heat stress on milk production in dairy cows. Heat stress reduces milk production and affects milk quality. Analyses of the fatty acid composition in the milk produced during summer display reduced concentrations of unsaturated fatty acids, especially long-chain unsaturated fatty acids. The authors performed RNA sequencing experiments and constructed six cDNA libraries from cow mammary tissue collected respectively during summer and winter. GO enrichment and KEGG pathway analysis showed that affected genes were mainly related to milk fat metabolism. Analysis of circRNAs revealed 19 upregulated and 19 downregulated circRNAs modulated in response to heat stress. Pearson’s test was used to establish the correlation of circRNA-mRNA and allowed identification of putative networks linking circRNAs-miRNAs to CD36 and/or SLC27. The results are interesting. However, the reported findings remain quite elusive and speculative and need to be validated by a series of experimental procedures. It is strongly recommended to include new data obtained with validation experiments of the identified molecular axes in the revised manuscript. Some additional files are either absent or impossible to open. Please double check.
Overall, the authors need to provide more complete and rigorous data to support their claims.
Answer: We really appreciate for your professional suggestion. At present, we are focusing on the validation experiments of the identified molecular axes, and any results will be published as soon as possible. And we have double checked the additional files in our computer, and they works well. Thanks so much.
Additional points are listed below
(1) Line 24-25. The reference number 21 is not appropriate. The manuscript from Poliseno and colleagues discloses a novel mechanism of PTEN regulation which is mediated by the sponge activity of PTENP1 toward specific microRNAs. CircRNAs are not mentioned in this paper.
Answer: Many thanks for your kind help, and we have replaced the reference 21 as “Li, X., L. Yang, and L. Chen. 2018. The Biogenesis, Functions, and Challenges of Circular RNAs. Molecular Cell 71(3): 428-442.”
(2) Line 49-50. Please include a relevant reference for the statement “In general, miRNAs play an important role in the processes of circRNA and mRNA interaction”. Why did the authors decide to focus on the putative role of microRNAs as mediators of the axis circRNAs-mRNAs? CircRNAs can act as protein sponges or even be translated into small peptides (Kristensen et al., 2019).
Answer: Thanks so much, and we have added a relevant reference in new line 169. Recent studies have showed that several abundant circRNAs can function as miRNA sponges, and we therefore focused on the molecular mechanism between circRNA and mRNA regulated by miRNAs. In the future, the functions of protein sponges and encoded peptides will be in our schedule.
(3)Line 13. The Authors should provide a graph related to the downregulation of CD36 and SLC27 after HS.
Answer: We have constructed circRNA-miRNA-CD36 (SLC27A6) networks by pairing the shared miRNA recognition sequences. And circRNA-miRNA-CD36 included 10 circRNAs and 16miRNAs, while circRNA-miRNA-SLC27A6 network only contained circCSN1S2 and miR-223. The results have been showed in Additional files.
(4)The expression levels of the validated circRNAs need to be assessed in the two experimental conditions used for the RNA-seq experiments (normal temperature and heat stress).
Answer: Yes, the expression levels of the validated circRNAs should be further assessed by RT-qPCR. In our paper, the mammary gland that obtained by percutaneous biopsies in stress group were prepared for RNA-seq in LC commary, and unfortunately no excessive mammary tissues were used for RT-qPCR analysis. And using the mammary gland obtained in winter group, we do a lot of additional work, such as RNase R treated experiment. The details were showed in Additional files 11 and Figure 4.
(5)The authors focused on the miRNA-sponging function of the identified circRNAs. The authors should ascertain the levels of the target miRNAs and their relative mRNA targets in the mammary glands in the two experimental conditions (normal temperature and heat stress), in order to validate the circRNA-miRNA-mRNA networks mentioned in the discussion.
Answer: we have made a plan to validate identified the circRNA-miRNA-mRNA networks, and the methodology that you suggested will give us a big help. Thanks so much for your kind help and suggestion.
Minor points:
(1) In Figure 1A/1B/1C, a detailed legend would help in distinguishing the two experimental conditions, by indicating which colors the histograms are referred to.
Answer: Thanks so much, and a color legend has been detailed in the note of Figure 1.
(2) Additional file 4 – impossible to open.
Answer: we do apologize for our mistake, and we have double-checked. It works well in my computer. Thanks so much.
(3) Additional file 5 and 6 are not present in the provided supplementary material.
Answer: we updated the Additional files in our supplementary material.
(4) Figure 2D should be positioned after figure 3B, coherently with the order of the main text.
Answer: we re-ordered the figres and tables in our main text. Thanks so much.
Round 2
Reviewer 1 Report
I would like to thank the authors for the effort made to answer my questions. I think that in this way I have been able to better understand the hypothesis of this work and part of the results.
Just two important comments:
a) I have not been able to find any of the figures mentioned in the text. Figure 1? 2 ? .... In the supplementary files some PDFs are included but I do not know if they have any kind of relationship. Please (the editor or the authors), either at the end of the text or as separate files I'd like to see the figures, it is difficult to understand all the text that supports the image without having the image itself.
b) About my previous question, number 5. It is not necessary to enrich the library for what I wanted to explain. In a miRNASeq, a size selection is made to eliminate the mRNAs, but in a "normal" RNASeq, those genes that encode miRNAS (average size around 90bp), can be detected correctly if they have a medium / high expression value.
Author Response
Comments and Suggestions for Authors
I would like to thank the authors for the effort made to answer my questions. I think that in this way I have been able to better understand the hypothesis of this work and part of the results.
Just two important comments:
(1) I have not been able to find any of the figures mentioned in the text. Figure 1? 2 ? .... In the supplementary files some PDFs are included, but I do not know if they have any kind of relationship. Please (the editor or the authors), either at the end of the text or as separate files I'd like to see the figures, it is difficult to understand all the text that supports the image without having the image itself.
Answer: Thanks so much for your comments. We submitted our figures as separate files in the past. At this time, we also added figures at the end of text.
(2) About my previous question, number 5. It is not necessary to enrich the library for what I wanted to explain. In a miRNASeq, a size selection is made to eliminate the mRNAs, but in a "normal" RNASeq, those genes that encode miRNAS (average size around 90bp), can be detected correctly if they have a medium / high expression value.
Answer: Many thanks for your professional suggestions. In our paper, we included libraries from other authors to check the expression of miRNAs. Based on your comments, we will try to check the expression of miRNA candidates by pre-miRNA with RNA-seq, and it is a good idea for our project.
Reviewer 2 Report
Unfortunately, without the requested data the paper is not supported.
Author Response
Comments and Suggestions for Authors
Unfortunately, without the requested data the paper is not supported.
Answer: Thanks so much, and we have provided a graph related to the downregulation of CD36 and SLC27 after HS. The details were showed in Additional file 9.
Round 3
Reviewer 1 Report
All my concerns were properly answered
Author Response
Thanks so much.
Reviewer 2 Report
As mentioned in the previous rounds of revision, the manuscript is interesting and well-designed. However, the reported findings remain quite elusive and speculative and need to be validated by a series of experimental procedures. It is strongly recommended to include new data obtained with validation experiments of the identified molecular axes in the revised manuscript.
The mentioned graph related to the down-regulation of CD36 and SLC27 after HS is not present in the additional file 9 provided. Please double check.
Author Response
Thans so much, and we have prepared and double checked new additional file 9 and additional file 11. They worked well. Additional file 9 showed the CircRNA-miRNA-mRNA correlation networks, while additional file 11 showed the expression analysis of circRNA candidates between different groups.
Round 4
Reviewer 2 Report
I still believe that the reported findings remain quite elusive and speculative without the requested validation experiments.
Since the identified molecular axes are not fully supported by the reported data I would suggest to attenuate the conclusions.